# Overcoming Barriers to Tobacco Cessation and Lung Cancer Screening among Racial and Ethnic Minority Groups and Underserved Patients in Academic Centers and Community Network Sites: The City of Hope Experience

**DOI:** 10.3390/jcm12041275

**Published:** 2023-02-06

**Authors:** Cary A. Presant, Kimlin Ashing, Dan Raz, Sophia Yeung, Brenda Gascon, Alexis Stewart, Jonjon Macalintal, Argelia Sandoval, Loretta Ehrunmwunsee, Tanyanika Phillips, Ravi Salgia, Amartej Merla, Shanmuga Subbiah, Michelle El-Hajjouie, Jeffrey Staley, Heather Graves, Ranjan Pathak, Shaira Dingal, Sagus Sampath, Beverly Laksana, Thomas Joseph, Tricia Eugenio, Veronica Degoma, Kathleen Burns, Sarah Phillips, Tingting Tan, Kelly Tarkshian, Virginia Sun, Arya Amini, Khristie Davy, Janet Cronkhite, Mary Cianfrocca, Susan Brown, Yuman Fong, Steven Rosen

**Affiliations:** City of Hope Medical Center, 1500 East Duarte Rd, Duarte, CA 91010, USA

**Keywords:** tobacco control, smoking cessation, cancer prevention, lung cancer screening, low dose Ct scans, LDCT, cancer disparities, minority health, personalized medicine, pathways to success

## Abstract

Background: Tobacco control is important for cancer patient health, but delivering effective low-dose CT (LDCT) screening and tobacco cessation is more difficult in underserved and patients from racial and ethnic minority groups. At City of Hope (COH), we have developed strategies to overcome barriers to the delivery of LDCT and tobacco cessation. Methods: We performed a needs assessment. New tobacco control program services were implemented focusing on patients from racial and ethnic minority groups. Innovations included Whole Person Care with motivational counseling, placing clinician and nurse champions at points of care, training module and leadership newsletters, and a patient-centric personalized medicine Personalized Pathways to Success (PPS) program. Results: Emphasis on patients from racial and ethnic minority groups was implemented by training cessation personnel and lung cancer control champions. LDCT increased. Tobacco use assessment increased and abstinence was 27.2%. The PPS pilot program achieved 47% engagement in cessation, with self-reported abstinence at 3 months of 38%, with both results slightly higher in patients from racial and ethnic minority groups than in Caucasian patients. Conclusions: Tobacco cessation barrier-focused innovations can result in increased lung cancer screening and tobacco cessation reach and effectiveness, especially among patients from racial and ethnic minority groups. The PPS program is promising as a personalized medicine patient-centric approach to cessation and lung cancer screening.

## 1. Introduction

Tobacco control is important for cancer patient health. Tobacco is the primary or contributing cause of 30% of cancers in the United States [1] and 80–90% of lung cancers are caused by or associated with tobacco use. Diagnosing lung cancer early can increase the 5-year cancer cure rate to 90.8% [2], emphasizing the benefit of effective implementation of low-dose CT (LDCT) for lung cancer screening (LCS). Since 5-year survival rates in lung cancer are 26% higher in patients undergoing tobacco cessation compared to patients who continue to use tobacco [3], implementing tobacco cessation programs is important as a fourth pillar of cancer care [4].

Delivering effective LCS and tobacco cessation is more difficult in underserved populations of patients from racial and ethnic minority groups [5,6]. This is likely due to poor access to care, insurance, language barriers, lack of trust, cultural beliefs or prejudices, and/or a lack of education about the benefits of tobacco cessation. At City of Hope (COH), we have developed strategies to overcome barriers to the delivery of LDCT and tobacco cessation across our network of academic and community sites. This communication summarizes our experience and recommendations.

## 2. Methods

We performed a needs assessment by conducting 193 interviews and surveys with clinicians, patients, tobacco treatment specialists, nurses, and administrators, in order to determine the barriers to tobacco control delivery. New tobacco control program services were implemented [7].

After the identification of a patient with current tobacco use, physicians were prompted to refer the patient to tobacco cessation. A multilingual tobacco treatment specialist (TTS) conducted a culturally sensitive motivational interview, gave educational materials, and referred the consenting patient for a tobacco cessation consultation. Patients with a qualifying tobacco use history were referred by the physicians themselves or with TTS prompting to LDCT screening. The use of multilingual support overcame a significant barrier to LCS in patients from racial and ethnic minority groups. Physicians considered patient life expectancy and willingness to have screening and potentially curative therapy before referral to LDCT screening.

We reviewed the tobacco use assessments and engagement with the tobacco control program by race and ethnicity across the City of Hope southern California treatment sites in 2021 (1 academic center and 40 community centers). We analyzed LCS rates and tobacco cessation referral rates, and cessation effectiveness after program implementation.

The COH tobacco control program consisted of quality improvement projects. This was submitted to the COH investigational review board, which concluded the program was deemed non-human-subject research. Therefore, no patient informed consent was required (IRB number 19201).

These new services were implemented in 2019 and continued until the present. Observational evaluations began in 2019 pre- and post-implementation.

## 3. Results

### 3.1. The City of Hope Catchment Area

In 2020, 137,125 new patients were seen in the entire enterprise of 1 academic center and 40 community sites in southern California (this does not include any patients seen in Cancer Treatment Centers of America in Chicago, Phoenix, and Atlanta, which were acquired by City of Hope in 2022, and does not include any new patients seen in the second academic center Lennar Hospital in Irvine, CA, USA, which opened in 2022). The details of these patients are presented in Table 1. In community centers, there were more patients with fewer Caucasians, Hispanics, and Asian/Pacific Islanders, and more Black, mixed, and other patients from racial and ethnic minority groups, mostly middle eastern and eastern European/Armenian. Across the entire enterprise, there were 51.4% of minority patients (39.8% in Duarte academic center and 59.5% in the community centers).

### 3.2. Barriers to Tobacco Cessation and City of Hope Solutions

Among patients, there was frequent refusals to participate in screening and/or cessation programs, especially among patients from racial and ethnic minority groups and patients of low socioeconomic status (SES). Stress in general or due to cancer therapies, anxiety and depression, work problems, and family problems contributed to patient reluctance. A lack of knowledge about tobacco-use-associated reduced cancer outcomes, increased side effects of cancer treatments, the high cost of cessation medications not covered by insurance, non-English speaking, and cultural obstacles all reduced patient agreement to initiate our cessation program, particularly in patients from racial and ethnic minority groups and low SES cancer patients. The cultural context was an important barrier for cessation staff to evaluate. For example, counseling was often stigmatized in Chinese patients because of the misinterpretation of cessation counseling as a mental health problem. Because our network of community clinical sites and our catchment area was so vast in southern California, travel to the academic center in Duarte was not convenient or possible for many tobacco users. Easy access to tobacco products particularly in low-SES neighborhoods and the use of tobacco products by family and friends served as ubiquitous triggers to continue tobacco use [8]. LDCT screening sites were more limited in low-SES areas.

In response to these observations in our needs assessment, City of Hope used implementation science to introduce solutions to improve tobacco-use screening and tobacco cessation [7], as outlined in Table 2.

Physician barriers were a lack of education about tobacco cessation, lack of time, and lack of reimbursement. These were addressed by educational training modules, newsletters from leadership, and improvements in the electronic medical record system.

Institutional barriers were the lack of leadership and personnel. These were addressed by hiring personnel (TTS full-time individuals), commitments from COH enterprise leaders, leadership newsletters published monthly, and naming physician and nurse champions (leaders) in academic center clinics and in network clinical sites.

To overcome patient barriers, the first key COH innovation was to include a Whole Person Care approach including training of tobacco treatment specialists to conduct motivational counseling of each patient, providing introductory tobacco use educational videos and brochures to patients even before asking for participation in individual cessation consultations, and TTS assistance to physicians to prompt LDCT screening of eligible patients.

A second COH innovation was to motivate clinicians to refer every patient with current tobacco use to the cessation program by leadership sending out a monthly newsletter (the Moonshot Shoutout) to every staff member reminding them about the importance of referral to cessation and ordering LDCT for eligible patients. This included training modules for physicians, nurses, and tobacco treatment specialists.

A third COH innovation was taking the promotion of cessation and screening to the patient point of care by training and deploying both clinician and nurse champions in selected clinics in the academic center and selected community centers [9]. A fourth COH innovation was providing patients with a choice of 36 distinct services from which a patient could select which services they initially wanted to use to begin their own personalized tobacco cessation and screening journey, which we named the Personalized Pathway to Success (PPS) program. This innovation was patient-centric and encouraged the continued engagement of the patient with the tobacco cessation specialist. A list of the services offered to patients in the PPS is seen in Table 3.

Specific tobacco control program and PPS program characteristics were introduced to target patients from racial and ethnic minority groups and/or underserved patients. Educational resources were multilingual, and the staff was diverse and representative of the patients from racial and ethnic minority groups to be emphasized. Monthly meetings with champions emphasized feedback about barriers encountered in screening and cessation. The use of video counseling allowed the inclusion of patients from racial and ethnic minority groups who were reluctant to proceed with face-to-face counseling. Offering family and friend support through the free tobacco-use support groups increased the trust of the patients in the program.

### 3.3. Teachable Moments for Referral to Tobacco Cessation and Lung Cancer Screening

Patients may be more agreeable to accepting referral to screening and tobacco cessation if they are approached at teachable moments. These include the visit at which the diagnosis of cancer is given to the patient; the visit at which the treatment plan is discussed with the patient (cessation is the fourth pillar of cancer care and should be in every treatment plan of a currently tobacco-using patient); a visit at which progression of the cancer is discussed with the patient; a preoperative visit for planning surgery; a visit for discussing cellular therapy with stem cell transplant or CAR-T cell therapy; and/or a visit for pulmonary consultation, pulmonary function testing, or respiratory therapy.

Meanwhile, tobacco cessation is a teachable moment for LCS. LCS is severely underutilized due to a variety of patient, provider, and system barriers [10,11]. The integration of LCS into tobacco cessation workflows is an important strategy for overcoming some of these barriers, including the identification of eligible participants and the lack of knowledge about LCS among people who use tobacco. At our institution, a comprehensive cancer center without affiliated primary care clinics, tobacco cessation services, and LCS are integrated, and the majority of patients screened for lung cancer are identified through referral for tobacco cessation [12]. We have trained tobacco cessation counselors to provide education about LCS, and there is one nurse practitioner and program coordinator for both programs. We have found that training tobacco cessation staff to provide LCS education is feasible and that tobacco cessation staff embrace providing LCS education as part of their role [13].

### 3.4. Results of Screening and Cessation Services

We have utilized several strategies to improve the utilization of LCS in our catchment areas, particularly in underserved communities with patients from racial and ethnic minority groups, rural populations, and non-English speakers. First, we have expanded our ability to provide LDCT to eligible patients by opening LCS programs at various community sites. Since the start of the COVID-19 pandemic, a face-to-face encounter for shared-decision making is no longer required, and telehealth visits have been expanded. This has allowed us to expand our screening program using a centralized staffing model from one site in 2019 to six sites throughout Los Angeles County and Orange County by 2023. Expanding the reach of our screening program helps to provide quality LCS to patients closer to their homes. We have also performed educational outreach to primary care providers in these areas, as well as in partnership with Federally Qualified Health Clinics (FQHCs) with large numbers of patients from racial and ethnic minority groups, resulting in improved utilization of LCS [14].

Educational outreach directed to patients about LCS, including community health fairs and educational material translated into Spanish, Chinese, Vietnamese, Korean, and Armenian languages has also been an important strategy.

Finally, we recently started an Early Lung Cancer Navigation program in the Antelope Valley, a community with the highest rates of lung cancer mortality and tobacco use in the Los Angeles area as well as large populations of African Americans, Latinos, and rural populations, to improve lung cancer screening and expeditious treatment of lung cancer. This program is led by a Community Navigator who has lived and worked in the Antelope Valley community, and the program is guided by a Community Advisory Board comprised of clinicians, lung cancer patients, and community stakeholders. The navigator performs a needs assessment of patients in our LCS program, patients with imaging findings that are suspicious for lung cancer, and patients with newly diagnosed lung cancer, and then helps guide the patient so that these potential barriers to care can be overcome. The program accesses funds for transportation and even internet-enabled tablets to allow patients easier access to see and communicate with their providers. In 2021, our program screened 153 new patients, which included 59% non-Hispanic Whites, 18% Hispanics, 12% Asians, 4% Blacks, and 7% other or declined to answer.

Tobacco cessation services have increased as a result of our expanded tobacco control initiatives after the needs assessment. Before the COH quality improvement project was implemented, documentation of Tobacco use was only 80.8%. Referrals of patients to tobacco cessation counselors were low at 1.4% of all patients. After the implementation of the project, the assessment of tobacco use increased to 96.6%. Referrals of patients increased to 6.2%.

Counseling of referred patients for consultation for cessation by a nurse practitioner increased to 98% in the Duarte academic center. At a 6-month follow-up of those patients, self-reported abstinence was 27.2%.

This was compared to the experience in the Antelope Valley community site (which had the highest tobacco use in the COH enterprise). Counseling by a TTS or consultation with a nurse practitioner increased to 83.2%. At a 6-month follow-up, self-reported abstinence was 22.5%.

The reach of this program was evaluated as part of the Cancer Center Cessation Initiative (C3I). Engagement of patients who used tobacco with at least one element of our tobacco treatment program was 93.0% of patients in the Duarte academic center and 59.6% of patients in our community site Antelope Valley with the highest smoking rate. The reach of participants measured by active participation in the tobacco treatment program was different for patients from minority groups 45.3%, compared to 66.6% of Caucasian patients.

LCS services have also increased as a result of our expanded tobacco control activities. Compared to the pre-expansion year of 2018, total LDCT referrals increased by 24.4%. Importantly, while referrals of Caucasian patients increased by 11%, referrals of patients from racial and ethnic minority groups increased by 59%.

To extend these previous results, we piloted the PPS program with tobacco treatment specialists and champions in the preoperative anesthesia testing clinic. Fifty-four patients were evaluated in the PPS project. We observed a 47% engagement of patients to initiate cessation. This was higher than the enterprise-wide historical engagement rate of counseling of only 6.2% before the cessation program innovations described above. The engagement rate of patients from racial and ethnic minority groups was slightly higher than that of Caucasian patients (not statistically significant). Self-reported abstinence from tobacco use (at 3 months after engaging in cessation) was achieved by 38% of patients and was slightly higher in patients from racial and ethnic minority groups than in Caucasian patients.

## 4. Discussion

Tobacco use is a major cause of human infirmity and reduced quality of life. Tobacco control is a major public health goal [15]. National guidelines for tobacco cessation [16] and LDCT screening [17] are widely accepted, but the effectiveness of tobacco cessation and LCS utilization remains low [18,19].

The population of patients from racial and ethnic minority groups in City of Hope is a large proportion of the enterprise’s cancer patients (51.4%), and the communication of methods and results of our interventions may be helpful to other institutions. Our focus has been on implementing personalized and patient-centric services, which have been actively supported by senior leadership and well accepted by clinical staff. Our program has been dependent on high staff resources and long-term support from the institution. Efforts must be continued to utilize methods that are likely to ensure the sustainability of the program [20] and attempt to minimize staffing needs [21,22].

It is remarkable that the pilot study results of the PPS cessation program achieved similar results in patients from racial and ethnic minority groups and Caucasian patients. If confirmed in our future studies, these preliminary results suggest that specific innovative strategies can possibly overcome barriers to tobacco cessation in patients from racial and ethnic minority groups, but broader implementation is needed to be certain of the success of these innovations. The observation that patients from racial and ethnic minority groups have higher tobacco usage does not necessarily imply that tobacco cessation is always more difficult, as our results with the PPS program suggest.

This experience was a quality improvement project. As such, it was observational and not a randomized research study. Thus, the methods used were based on combining separate principles that had been developed using implementation science at COH and other institutions. Our project showed that using these principles with trained multilingual support personnel can result in valuable benefits for both patients from racial and ethnic minority groups and Caucasian patients. The PPS pilot study is extending these principles in a novel fashion to academic and community sites, focusing on teachable moments in the cancer patient experience.

Limitations of this communication include that this was performed in only one institutional enterprise limited to southern California and should be extended to other institutions and locations. The PPS program was a pilot, and an extension to other clinics and centers is in progress. The number of patients in the pilot PPS program is low and larger numbers will be required to determine specific services that are most effectively utilized by specific patients from racial and ethnic minority groups.

## 5. Conclusions

Using innovative strategies and implementation science, lung cancer screening, and tobacco cessation programs can improve outcomes and may reduce disparities in patients from racial and ethnic minority groups.

## Figures and Tables

**Table 1 jcm-12-01275-t001:** Characteristics of City of Hope Patients 2020.

	Duarte	Community Centers
Number of New Patients	51,099	86,026
% Tobacco Users	4.5%	7.4%
Race and Ethnicity		
Caucasian	60.2%	40.5%
Hispanic	21.9%	16.2%
Asian, Pacific Islander	8.2%	5.4%
Black, African American	8.8%	29.7%
Mixed, Other	0.9%	8.2%

**Table 2 jcm-12-01275-t002:** Barriers and City of Hope Solutions for Tobacco Cessation and LDCT Screening.

Barrier	City of Hope Solution
Initial patient refusal to consider cessation	Motivational interviewing by TTS TTS Offers PPS Involve champion leader or attending oncologist in Duarte clinic or community center
Lack of knowledge about the risks of tobacco use and benefits of screening and cessation	Educational videos and brochures
Lack of LDCT access in low SES areas	Transportation to Duarte LDCT Collaborate with local hospitals to provide LDCT City of Hope builds LDCT services in community sites
Tobacco use triggers are prevalent with family, friends, and community smoke shops	Consultation and counseling about triggers Free Tobacco use support groups for patients, family, and friends Referrals to psychological counseling Collaborate with local city councils about reducing access to Tobacco use
Cultural barriers and lack of trust in Caucasian cessation providers	Staff program with TTS and clinicians representing patients from racial and ethnic minority groups (Hispanic, Chinese, Filipino, Black, Middle Eastern/Armenian) Training staff in cultural and societal values in patients from racial and ethnic minority groups
Non-English-speaking patients	Staff with bilingual TTS Online or in-person translation services
High cost of cessation medications	Pharmacists assist in finding affordable medications Availability of charity funds
Inconvenient or lack of travel to face-to-face consultations or LDCT screening	Telehealth services for consultations and counseling Transportation services by local and community service providers
Lack of referral by clinicians	Newsletter from leadership Opt out, opt in, and Best Practice Advisory referrals in electronic medical records
Lack of reimbursement for clinical cessation services	Train clinicians in using CPT codes 99406 and 99407 Provide “smart-phrase” to assist documentation of cessation services Up-code E/M code selection appropriately if cessation services provided For managed care, refer to the contracted in-network cessation provider

Abbreviations: TTS: Tobacco treatment specialist; LDCT: Low-dose computerized tomography; PPS: Personalized pathway for success; CPT: current procedural terminology.

**Table 3 jcm-12-01275-t003:** Menu of Personalized Pathway to Success (PPS) Services.

Introductory Services
Motivational interview
“Tobacco Cessation as recovery enhancement” video
Tobacco risks and cessation benefits brochure
**Essential Elements**
Consultation with Tobacco Cessation provider
Nicotine replacement therapy (NRT) prescription
Varenicline or Bupropion eligibility evaluation and prescription
Combination NRT and oral cessation medication prescription and
Initial Tobacco Use Assessment for type of tobacco and doses
City of Hope (COH) tobacco use support group intake form and registration
Individual counseling and support by the Tobacco Treatment Specialist
Rapid action plan for relapse or slips
Educational video, brochure and counseling on preventing tobacco use
Follow up Tobacco Use Assessments
**Support Resources**
Identify and recruit a “buddy” support friend or a participant peer from
Identify and recruit an “angel” support family member
Offer non-COH support groups: Nicotine Anonymous, Celebrate
Phone support: Kick It CA Quitline
Phone Apps: SmokeFree, QuitGuide, QuitStart
Text Support: SmokeFree Text, Kick It CA text, “DITCHJUUL”
Online live chat: kickitca.org, cancer.gov
Web Resources: smokefree.com, becomeanex.org, trughinitiative.org,
**Commitment Adherence Tools**
Commitment Agreement
Quit Plan: https://smokefree.gov/build-your-quit-plan (accessed 31 January 2023)
**Daily Coping Strategies**
5Ds: Distraction, Delaying, Drinking Water, Deep Breathing, and
7 self care behaviors: health coping, healthy eating, regular exercise,
Cinnamon stick/bubble blowing
COH Relaxation Video with Guided Imagery:
COH tobacco Cessation Hypnosis Video: Hypnosis for Smoking Cessation-You Tube (accessed 31 January 2023)
Audiobooks accessible through *Overdrive* and *Audiobooks* apps
Stress management apps: *Headspace* and *Insight Timer*
Spiritual care and support
**Educational Tools**
Educational handouts
Kick It https://www.youtube.com/c/KickItCa/videos (accessed 31 January 2023)
The CDC E-Cig cessation factsheet
Modular videos
NCCN information sheets

Abbreviations: CDC: Center for Disease Control; E-Cig: electronic cigarettes; NCCN: National Comprehensive Center Network.

## Data Availability

The data are available for review at City of Hope Medical Center, Department of Population Science.

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
