# Peer review of "Overcoming Barriers to Tobacco Cessation and Lung Cancer Screening among Racial and Ethnic Minority Groups and Underserved Patients in Academic Centers and Community Network Sites: The City of Hope Experience"

_jcm, 2023, doi:10.3390/jcm12041275_

Round 1
Reviewer 1 Report
Thank you for the opportunity to review this paper. The goal of this manuscript was to describe the strategies developed to overcome barriers to the delivery of low dose computed tomography and tobacco cessation at City of Hope. This a well-written paper that provides information on how smoking cessation and lung screening may be integrated into cancer care settings. I have included comments for consideration below. I also have one overarching comment/question that requires attention throughout the manuscript.
It is my understanding these new strategies were implemented in the City of Hope cancer care setting such that they were interacting with patients with a current cancer diagnosis. While it is well understood the importance of smoking cessation among people with a cancer diagnosis, additional attention is needed on the importance of prioritizing lung cancer screening in this setting and patient population as well. Specifically, among those who were identified as eligible for lung screening was special consideration given to the patient’s life expectancy or the ability or willingness to have curative lung surgery?
Introduction –
Lines 34-35, since the focus of this paper is on tobacco and lung cancer, it may be helpful to add to the introduction that 80-90% of lung cancers are due to tobacco use.
Lines 37-39, please clarify if 5 year survival rates are 26% higher among patients who do not smoke vs those who smoke or if it is higher among those who quit smoking vs those who continue to smoke. The term ‘undergoing tobacco cessation’ is confusing as currently written.
Lines 40-41, if there is space it would be helpful to cite some of the reasons why delivering effective LCS and tobacco cessation is more difficult in underserved and minority populations.
Consider removing the term 'smokers' throughout the manuscript, replacing it with people first language.
The authors may also wish to consider using the term ‘minoritized’ as recommended here: https://jamanetwork.com/journals/jama/fullarticle/2783090
Methods –
Please provide additional details regarding the implementation period and the evaluation period. Please indicate when the new program services were implemented and if these new strategies were evaluated in an ongoing manner, pre-post-implementation, or post-implementation only.
Lines 45-46, How many interviews were conducted as part of the needs assessment?
Results –
In the methods, the authors mention interviews were conducted with several stakeholders (e.g., patients, nurses, clinicians), however, the barriers summarized seem to come from only the patient perspective. If additional barriers were identified by other stakeholders, they should be mentioned here.
Line 124, replace ‘barrier’ with ‘barriers’.
Lines 178-179, how was abstinence from tobacco use assessed in this effort? Was it done via self-report or biochemical verification?
Discussion –
Lines 202-203, it is mentioned that the number of patients in the pilot PPS program is low, but it is unclear how many patients participated in this pilot. Earlier in the manuscript, it states that 153 new patients were screened and that 137,125 new patients were seen in the COH catchment area. Please clarify how many patients were reached and who engaged in the pilot PPS program. These sample sizes should be integrated earlier on in the paper.
Author Response
We have inserted language at original line 47 to indicate that referrals to lung cancer screening considered patient life expectancy and willingness to have lung cancer treatment.
We have added language at original line 35 to indicate that 80-90% of lung cancers are related to tobacco.
We have clarified the text at original line 38 to indicate that the higher survival rate is associated with tobacco cessation compared to patients who continue smoking.
We have added text at original line 41 to indicate barriers for LCS and cessation in underserved and minority populations.
We have changed the term smokers throughout and changed it to tobacco users.
We have changed the term minorities throughout to racial and ethnic minority groups.
In methods at original line 47, we have added text to indicate when the new strategies were implemented and that new strategies were evaluated pre and post implementation.
In original line 45 we added how many interviews were conducted as part of the needs assessment.
In the methods in original line 84 we added text to identify barriers in other stakeholders.
In original line 124, we replaced barrier with barriers.
After original line 166 we have added a paragraph on the reach of the program.
In original line 178, we added text to indicate that abstinence was evaluated by self report.
At original line 174, we added the number of patients in the PPS project.
We added the number of patients reached and who engaged in the PPS program in original line 172.
We added a paragraph describing the increase in lung cancer screening services in original line 180.

Reviewer 2 Report
The manuscript deals with the role of innovative strategies to promote lung cancer screening and associated tobacco cessation programs. The material comprises of 137,125 new patients in South California at City of Hope catchment area in its academic center and 40 community centers.
The value of early low dose CT for lung cancer screening is shown in the cited study (2) and that of associated smoking cessation in studies (3,4). The manuscript emphasizes the problems among underserved and minority populations. Study 5 shows the potential barriers among the minority populations. The fact that those populations have higher smoking rates than others does not necessarily mean that practical and personal smoking cessation intervention would be more difficult.
The overall number of patients is great, and their ethnic background is shown, as is the information that some 5-6 per cent of them were smokers. Thus, the number of smokers was somewhat below 8000. It would been interesting to learn a little more about the practical arrangements of the screening and the associated smoking cessation arrangements.
The text in the result part describes the material, the developed solutions to overcome the barriers and discussion on “teachable moment”. All this is interesting but to me more methods or background. The actual results part is the 3.4. “Results of Screening sand Cessation Services”.
A reader of a scientific paper is obviously interested in the actual results and the facts to justify them. In this respect the manuscript is very vague. The actual data is not given, only some numbers that are not easy to assess without proper methods and data information.
There is little doubt that the developed many methods to overcome the barriers ae valuable, but the study does not as such test those methods. The results only indicate that combining many good principles with practical intensive work even in disadvantaged populations can bring good sand valuable effects.
Author Response
We have added language at the original line 199 to add that higher smoking cessation rates in minority groups does not necessarily mean that cessation intervention would be more difficult.
We have added language in the methods section in original line 47 to describe the workflow for screening and cessation.
More methods description has been added in original line 47 to improve the descriptions of our solutions.
More data has been given in original line 164 and in lines 166-171 for clarity and completeness of our results.
We have added a description in the methods section stressing that this was a quality improvement project and was not human research, as indicated by our IRB.
We have added language in the discussion indicating that combining many good principles can produce valuable results even in disadvantaged populations.
We have inserted language at original line 47 to indicate that referrals to lung cancer screening considered patient life expectancy and willingness to have lung cancer treatment.
We have added language at original line 35 to indicate that 80-90% of lung cancers are related to tobacco.
We have clarified the text at original line 38 to indicate that the higher survival rate is associated with tobacco cessation compared to patients who continue smoking.
We have added text at original line 41 to indicate barriers for LCS and cessation in underserved and minority populations.
We have changed the term smokers throughout and changed it to tobacco users.
We have changed the term minorities throughout to racial and ethnic minority groups.
In methods at original line 47, we have added text to indicate when the new strategies were implemented and that new strategies were evaluated pre and post implementation.
In original line 45 we added how many interviews were conducted as part of the needs assessment.
In the methods in original line 84 we added text to identify barriers in other stakeholders.
In original line 124, we replaced barrier with barriers.
After original line 166 we have added a paragraph on the reach of the program.
In original line 178, we added text to indicate that abstinence was evaluated by self report.
At original line 174, we added the number of patients in the PPS project.
We added the number of patients reached and who engaged in the PPS program in original line 172.
We added a paragraph describing the increase in lung cancer screening services in original line 180.